# Utilizing Amino Acid Composition and Entropy of Potential Open Reading Frames to Identify Protein-Coding Genes

**DOI:** 10.3390/microorganisms9010129

**Published:** 2021-01-08

**Authors:** Katelyn McNair, Carol L. Ecale Zhou, Brian Souza, Stephanie Malfatti, Robert A. Edwards

**Affiliations:** 1Computational Sciences Research Center, San Diego State University, 5500 Campanile Drive, San Diego, CA 92182, USA; 2Lawrence Livermore National Laboratory, Global Security Computing Applications, Livermore, CA 94550, USA; zhou4@llnl.gov; 3Lawrence Livermore National Laboratory, Biological Sciences Research Division, Livermore, CA 94550, USA; souza21@llnl.gov (B.S.); malfatti3@llnl.gov (S.M.); 4College of Science and Engineering, Flinders University, Bedford Park, SA 5042, Australia

**Keywords:** phage, genome, gene, annotation, machine learning, clustering, prediction

## Abstract

One of the main steps in gene-finding in prokaryotes is determining which open reading frames encode for a protein, and which occur by chance alone. There are many different methods to differentiate the two; the most prevalent approach is using shared homology with a database of known genes. This method presents many pitfalls, most notably the catch that you only find genes that you have seen before. The four most popular prokaryotic gene-prediction programs (GeneMark, Glimmer, Prodigal, Phanotate) all use a protein-coding training model to predict protein-coding genes, with the latter three allowing for the training model to be created ab initio from the input genome. Different methods are available for creating the training model, and to increase the accuracy of such tools, we present here GOODORFS, a method for identifying protein-coding genes within a set of all possible open reading frames (ORFS). Our workflow begins with taking the amino acid frequencies of each ORF, calculating an entropy density profile (EDP), using KMeans to cluster the EDPs, and then selecting the cluster with the lowest variation as the coding ORFs. To test the efficacy of our method, we ran GOODORFS on 14,179 annotated phage genomes, and compared our results to the initial training-set creation step of four other similar methods (Glimmer, MED2, PHANOTATE, Prodigal). We found that GOODORFS was the most accurate (0.94) and had the best F1-score (0.85), while Glimmer had the highest precision (0.92) and PHANOTATE had the highest recall (0.96).

## 1. Introduction

The first genome ever sequenced was that of Bacteriophage MS2 [1]. Twenty years later, the first bacterial genome, *Haemophilus influenza*, was sequenced, and with it came the need to computationally predict where protein-coding genes occur in prokaryotic genomes [2]. This gave rise to the first of the gene annotations tools, GeneMark [3], Glimmer [4], and CRITICA [5], a decade later Prodigal [6], and most recently PHANOTATE [7] for viral genomes. One thing each of these tools shares in common is the necessity for a training set of good genes—genes that are highly likely to encode proteins, and that the software can use to learn the features that segregate coding open-reading frames (ORFs) from noncoding ones. GeneMark and GLIMMER, and to an extent Prodigal and PHANOTATE, all require precomputed gene models to find similar genes within the input genome, and the better these training models, the better the predictions that each tool makes. Both GeneMark and CRITICA rely on previously annotated genomes to predict genes in the input query genome. GeneMark selects one of its precomputed general heuristic models based on the amino acid translation table and GC content of the input genome. CRITICA uses the shared homology between the known genes and the input genome, as well as non-comparative information such as contextual hexanucleotide frequency. In contrast, GLIMMER, Prodigal, and PHANOTATE create gene models from only the input query genome, which removes the dependence on reference data. Each uses a different method to select ORFs for inclusion in a training-set, on which a gene model is built. GLIMMER builds its training-set from the longest (and thus most likely to be protein-encoding) ORFs, which are predicted by its LONGORFS program. Prodigal uses the GC frame plot consensus of the ORFs, and performs the first of its two dynamic programming steps to build a training-set. PHANOTATE creates a training-set by taking all the ORFs that begin with the most common start codon ATG.

An additional method, which has subsequently been added as an option to LONGORFS, is the Multivariate Entropy Distance (MED2) algorithm [8], which finds the entropy density profiles (EDP) of ORFs and compares them to precomputed reference EDP profiles for coding and noncoding ORFs. The EDP of an ORF is a 20-dimensional vector *S* = {*s_i_*} of the entropy of the 20 amino acid frequencies *p_i_*, and is defined by:(1)$$\;s_{i}=\frac{-1}{H}p_{i}logp_{i}\;\;\;\mathrm{where}\;\;\;\;H=-\sum_{j=1}^{20}p_{j}log\;p_{j}$$

This approach relies on the coding ORFs having a conserved amino acid composition that is different from the noncoding ORFs. This differential can be seen when comparing the observed amino acid frequency of known phage protein coding genes to the expected frequencies (Figure 1A), which are based purely on the percent AT|GC content of the genome. Since coding ORFs have a bias towards certain amino acids, and noncoding ORFs have frequencies approximately dependent on the nucleotide composition, each will cluster separately in a 20-dimensional amino acid space (Figure 1B).

MED2 uses these observed amino acid frequency EDPs from known coding/noncoding genes, in the form of average “centers”, to classify potential ORFs in other genomes. One problem that arises is that the observed amino acid frequencies, and the EDPs derived from them, change based on the AT/GC content of a genome (Figure 1B). This is because the codon triplets for amino acids do not change even when the probability of those codons occurring changes due to AT/GC content. The way MED2 overcomes this is by using two different sets of coding/noncoding references, one for low-GC content genomes and one for normal and high-GC content genomes. Aside from the complication of forcing a discrete scale onto continuous data, there is also the issue that the amino acid composition can change independent of GC content. A good example of this is the *Propionibacterium* phages, which are the darker orange points at (4,1) in Figure 1B that cluster with the yellow high-GC content genomes despite having a normal GC content. Another problem with reference-based gene prediction, and the pitfall of all supervised learning, is that only genes similar to those already known are predicted. As such, we sought to implement a reference-free method for identifying protein-coding genes within a stretch of DNA, herein titled GOODORFS. Our approach takes the EDP metric, and expands on it by adding the three stop codons and the ORF length. However, rather than using reference EDP profiles, we use unsupervised learning (namely KMeans) to cluster the ORFs, and then denote the cluster with the lowest variation as the coding ORFs.

## 2. Materials and Methods

The GOODORFS program, along with the data, can be obtained from the GitHub repository (http://github.com/deprekate/goodorfs). Currently, GOODORFS is available as Python3 code with the dependencies NUMPY [9] and scikit-learn [10]. PCA transformations were performed using the decomposition package from scikit-learn, and plots were created using Matplotlib [11]. Points were plotted in an interlaced pattern based on either the stop codon location or the genome ID, rather than by categorical legend order. The alpha transparency was adjusted to suit individual plots.

### 2.1. Data

A list of 14,254 phage genome IDs was downloaded from the October 2019 snapshot of the Millard Lab bacteriophage database [12], and subsequently the corresponding FASTA files were retrieved from GenBank. The genome sizes range from 1417 bp to 497,513 bp, with a mean length of 58,497 bp ± 53,767 bp. From this dataset, we removed 55 genomes that came back empty, and 20 that belong to Mollicutes phages (9 Spiroplasma, 8 Mycoplasma, 3 Acholeplasma) that use an alternative codon translation table, leaving us with a total of 14,179 phage genomes. Since genome annotations in GenBank are quite often incomplete or incorrect, we reannotated these genomes using the four available annotation programs (GeneMarkS, version 4.32; Glimmer3, version 3.02; Prodigal, version 2.6.3; and PHANOTATE, version 1.5.0), and to ensure reliable predictions, we only included predicted genes that were called by two or more programs. GeneMarkS was run with default parameters and the --*phage* option. Glimmer was run using the supplied *g3-from-scratch* script, which creates a gene model from the input genome sequences using the LONGORFS program. For Glimmer, we changed the default minimum gene length from 90 bp to 87 bp, to match the other three programs’ gene length default settings. This is because Glimmer does not include the three nucleotides of the stop codon in the calculation of gene length, unlike the three other tools. Prodigal was run with default parameters, with a change to the source code lowering the MIN_SINGLE_GENOME value from 20,000 to 1, to allow it to run on genomes smaller than 20 k bases. PHANOTATE was used with all default settings.

### 2.2. Algorithm

The GOODORFS algorithm, shown in Figure 2, is comprised of four main steps: finding the ORFS, calculating the ORF EDPs, clustering, and choosing the cluster that contains the coding ORFs.

#### 2.2.1. Finding ORFS

For each genome, all potential ORFs were found in both frames by finding any start codon (ATG, GTG, TTG) and then following it to a stop codon (TGA, TAA, TAG), and only taking ORFs with lengths equal to or greater than 90 bp. We also include ORFs that run off either end without requiring either a start or stop codon, setting the start/stop position to the first/last available codon. This usage of multiple start codons per single stop codon leads to essentially having the “*same*” ORF, just alternate truncated versions, in our data. This was done because the correct start codon was not known, but the assumption was made that during clustering, the ORFs with correct start positions will cluster separately from those with incorrect start positions. We do, however, limit the number of these “redundant” ORFs in the data, by only including the first half (going from the outermost/furthest start codon in towards the stop codon) of the alternate start position’s ORFs. This was because some ORFs would have hundreds of alternate start positions, whether due to extremely long genes (as in the case of tape-measure proteins or RNA polymerase, which can be thousands of amino acids long), or shorter genes that have a repeated motif that includes start codons. Additionally, sequencing or assembly errors, where the same codon is repeated hundreds of times, can create an ORF that is composed entirely of start codons. Each of these alternate start positions all belong to the *same* stop codon; therefore, the term “*unique*” ORF can be thought of as one specified by a single unique stop codon, and thus, all alternate start positions are essentially the same ORF.

#### 2.2.2. Calculating the EDPs

Each ORF was translated using the Standard Code (GenBank’s translation Table 1) [13] into the 20 amino acids (ARNDCEQGHILKMFPSTWYV) and the 3 stop codons amber (+), ochre (#), umber (*). The 23 characters were counted and then divided by the total to get the frequencies, and the EDP was calculated according to Equation (1). To the EDP vector we also added a 24th element, the length of the ORF. Before clustering, we normalized the features of our data using the StandardScaler function from scikit-learn.

#### 2.2.3. Clustering

Clustering was performed using the KMeans function (default parameters) from scikit-learn with three clusters for genomes with less than 450 unique ORFs, and four clusters for those with more. In order to control for the stochasticity of KMeans, which is not a deterministic algorithm, one thousand models were created, and the model with the lowest inertia was kept.

#### 2.2.4. Choosing a Cluster

To predict which cluster contained the coding ORFs, we calculated the mean absolute deviation (MAD) of the points, excluding the ORF length values, summing the remaining 23 elements, and selected the one with the lowest value. The ORF length feature is excluded because even though it helps to resolve clusters, the coding ORFs tend to have a larger variance in their lengths, which biases the MAD sums. Even by limiting the number of alternate start sites per ORF included into the data, quite often clustering would assign all of these same ORFs to their own cluster. To overcome this, if a cluster was composed of less than 5% of the total unique ORFs in the dataset, that cluster and the next lowest MAD cluster were merged and selected as the coding cluster. This value was chosen because without this cutoff there were 193 genomes that failed (F1 < 0.1), and we noticed that many of these had an unusually low percent of unique ORFs. The mean percent of unique ORFs was 4.0%, which we rounded up to 5%. From the ORFs of the predicted coding cluster, we took the longest alternate truncation for each unique ORF and added it to the output set of good ORFs.

### 2.3. Performance Analysis

The MED2 program was run with all the default values except with a change in the source code of minimum ORF length from 90 bp to 87 bp, because like Glimmer, it does not include the stop codon triplet in the ORF length calculation. We only ran the MED2 algorithm, and not the Translation Initiation Sites (TISModel) program, which uses Ribosomal binding site motifs to further refine the MED2 ORF predictions. The LONGORFS program was run during the Glimmer consensus annotation step, using the parameters above. Both Prodigal and PHANOTATE do not have functionality to log the set of training genes created, so a single line was added to the source code of each repo to print out the stop codon (which is a unique identifier) of each gene in the training set. The *diff* patch files to make this change are available in the GOODORFS repo, and can be applied via the command *git apply*.

## 3. Results and Discussion

We began our work by taking the previously published EDP metric, but expanded it to also include the three stop codons, amber (+), ochre (#), and umber (*). We also appended the length of the ORF to the vector. The purpose of this was two-fold: First, the EDP metric loses informational content when converting from amino acid counts to frequencies. Certainly, a short ORF with a given frequency coding bias is less significant than a much larger ORF with the same frequency bias, since the latter maintains that bias over many more codons. Second, the ORF length bolsters the clustering step, since coding ORFs are generally longer than noncoding ones (for our dataset, the mean lengths were 595 bp and 345 bp, respectively).

To demonstrate our approach, we took the representative genome, *Caulobacter* phage CcrBL9 [14], chosen since it is the largest genome in our dataset that has a high GC content (>60%), so it will have significantly more noncoding ORFs than coding, which allows for visualizing all of the categories in Figure 3. We then found all the potential ORFs in the genome, calculated their 24-dimensional EDPs, and used PCA analysis to plot them in 2 dimensions (Figure 3A). The coding ORFs (blue) and the noncoding ORFs (red) cluster separately, and this same pattern is observed across all other genomes.

Initially we began using KMeans with two clusters, coding and noncoding. This worked well for genomes with average and low GC contents, but did not work well on high-GC content genomes (mean F1-scores were 0.72 and 0.49, respectively). This is because the probability of encountering a stop codon is lower in those with a high GC content, and so they have about twice as many noncoding ORFs as coding (Appendix A). This can be seen in Figure 3A, where the red noncoding ORFs far outnumber the blue coding ORFs. Because the KMeans algorithm tends to distribute points into equally sized clusters, this would lead to far more noncoding ORFs being incorrectly clustered with the coding ORFs. To overcome this, we initially changed to the KMeans of three clusters, so that the coding ORFs would get their own cluster and the noncoding ORFs would be split between the two others clusters. Upon inspection of the data, it is apparent that the noncoding ORFs fall into six different clusters, one for each reading frame offset (Figure 3B), and that not all ORFs have completely random non-conserved amino acid frequencies. The first cluster of noncoding ORFs are those in the intergenic (IG) regions, and for the most part, these have amino acid frequencies that are completely random and follow the expected frequency based on GC content alone. The other five clusters are those that overlap with a coding ORF, and are denoted by an offset, which can be thought of as how many base shifts it takes to get to the coding frame. In contrast to the IG, the offset clusters overlap with a coding ORF, and have a slightly conserved amino acid frequency. This is because even though they themselves do not encode proteins, they are not independent of the coding ORFs they overlap with; they share the same nucleotides, just offset and in different frames. These non-intergenic noncoding ORFs fall into five categories (1+, 2+, 0−, 1−, 2−), and are always in relation to the coding frame, denoted as 0+ (which is synonymous with coding). The same-strand categories (1+ and 2+) correspond to when the coding ORF is in frame *n* and so the two noncoding ORFs are n + 1 and n + 2. Likewise, the three opposite-strand categories (0−, 1−, and 2−) correspond to a coding ORF at frame n and noncoding ORFs at n + 0, n + 1, and n + 2, except in the reverse direction. Thus, we would expect there to be one conserved amino acid cluster for the coding frame, five spurious semi-conserved clusters, and one cluster for intergenic regions (Figure 3B and Appendix A). Since we are working with phages, which can range down to only 18 unique ORFs for a genome (Appendix A), it is not possible to cluster ORFs into these seven clusters, so we settle on clustering into three for small genomes (less than 450 unique ORFs) and four clusters for larger genomes (over 450 unique ORFs). This cutoff was chosen because when using four clusters, genomes with less than 400 unique ORFs started to fail. Likewise, when using only three clusters, genomes with more than 500 unique ORFs began failing. We could have continued this pattern of setting multiple staggered cutoffs depending on genome size, until reaching seven clusters, but did not observe significant improvements when using more clusters, even for the largest genomes. However, we still need to pick which cluster contains the coding ORFs, since unsupervised clustering does not assign labels. Since coding ORFs have a conserved amino acid frequency, they will have a higher “density” cluster when compared to noncoding ORFs—in 24-dimensional ordination the points will be highly clustered. This can be observed in Figure 3B, where the blue coding ORFs are a dense cluster, while the noncoding ORFs are a sparse cloud. We used the mean absolute density (MAD) of each cluster to quantify the “density” in the 23-dimensional EDP space, since the MAD better accounts for outliers in the data. The cluster with the lowest sum of MADs (excluding the ORF length feature) was selected as the coding cluster.

To test the efficacy of our method, we took a set of 14,179 annotated phage genomes, and for each genome we identified all potential ORFs, calculated the amino acid EDPs, clustered them, labeled the densest cluster as coding, and then calculated the F1-score to measure the performance. The average of the F1-scores was 0.85 ± 0.13, with only 32 genomes failing (an F1-score of less than 0.1). Unsurprisingly, all the failed genomes were very small in size, with ten or less annotated protein-coding genes (and 60 or less *unique* ORFs). A breakdown of the genome sizes in our dataset (both number of protein-coding genes and all potential ORFs) is shown in Appendix A. Additionally, all the genomes that failed belonged to the taxon *Microviridae*. Whether this is due to correlation (about 62% of the genomes with less than 60 unique ORFs were *Microviridae*) or causation remains undetermined. As shown in the plot of amino acid versus GC content (Figure 1B), there is a large cloud of points that do not follow the horizontal trend, and instead cluster at the top of the figure separately along the vectors that represent the FSQC amino acids, where most of the genomes belong to *Microviridae* (Appendix A). This suggests that the *Microviridae* do not use the Standard Code, but rather one that potentially substitutes one or more of the over-observed amino acid codons (FSQC) for the codons that are under-observed (VDE). This hypothesis is supported by the dozen or so phages from other taxa that group with the *Microviridae* away from their expected locations in Appendix A. Examples of two non Microviridae phages with unusual amino acid composition are shown in Figure 4, where no discernable separation of coding from noncoding is observed. Both of these genomes cluster away from the expected frequency with the Microviridae (Appendix A), and the larger genome sizes reinforces the possibility that they are using a different codon translation table.

We compared the F1-scores of our analysis to those from four other similar programs that create a training set of genes from an input genome: LONGORFS (Glimmer), MED2, PHANOTATE, and Prodigal (for PHANOTATE and Prodigal, only the initial training set creation step is run, and not the entire gene-finding algorithm). The LONGORFS program had 49 genomes had an F1-score less than 0.1, and the overall mean F1-score was 0.53. For MED2, there were six genomes that caused the program to crash; of the remainder, there were 1143 genomes with an F1-score less than 0.1, and the mean F1-score was 0.55. No genomes had an F1-score less than 0.1 with PHANOTATE, since its method sacrifices precision for recall, and subsequently PHANOTATE had the lowest mean F1-score of 0.43. Prodigal performs well, with an F1-score of 0.75 and no genomes failing (an F1-score less than 0.1).

Comparing the mean F1-scores of the four alternative methods to the 0.83 obtained from GOODORFS shows just how much better our method is at recovering coding ORFs compared to other methods (Table 1). Of the four alternative methods, Prodigal performs the best, but at this point in the complex algorithm, Prodigal has already performed one of its two rounds of dynamic programming to identify coding ORFs. Plotting the individual genome F1-scores from GOODORFS versus the four other methods shows the distribution (Figure 5), wherein points above the diagonal identity line, of which there are many, signify that GOODORFS is outperforming the competing method. Many of the points in Figure 5 that have low F1-scores (<0.5) with GOODORFs also happen to belong to the class of genomes that have unusual amino acid frequencies. The two representative genomes discussed above, *Ralstonia* phage RSM1 and *Escherichia* phage fp01, had GOODORFS F1-scores of 0.38 and 0.23, respectively. These two genomes did not fare much better with the other programs with the highest F1-scores of 0.56 and 0.41 coming from Prodigal, which is not beholden to codon translation tables, except in the form of start and stop codons.

All the previous performance results are based on simple ORF counts. If we were to instead normalize by ORF length, the F1-scores for methods that favor recall over precision (GOODORFS, Phanotate, Prodigal) would all increase because, as we have shown, the true-positives (coding ORFs) are longer than the false-positives (noncoding ORFs). This is relevant because training on a gene set usually involves iterating over the length of the genes, and so longer ORFs have more emphasis than shorter ORFs. An example of this would be in calculating synonymous codon usage bias in a set of genes, where a long ORF with many codons would contribute more than a short one with few codons. Another aspect of training on a gene model is that a high F1-scoring method might not be the best choice for a given gene prediction tool. Sometimes it is better to have a very precise training set with very few false-positives. For instance, this might be the case with Glimmer, which uses the very low F1-scoring method LONGORFS to create a training set. Perhaps having only a few true-positive genes is worth not having any false-positives contaminating the training gene model. Whether to adopt our GOODORFs method into phage gene-finding tools or pipelines is left up to the researchers to determine on a case-by-case basis.

## 4. Conclusions

We have demonstrated the benefits of our GOODORFS method over similar alternative methods. The more accurate the set of genes used for training a gene model, the more accurate the final gene predictions will be, and this will lead to significant improvements in current gene-finding programs. We plan on incorporating GOODORFS into the PHANOTATE code to replace its current naïve method for creating a training set of all ORFs that start with the codon ATG. Glimmer, which uses LONGORFS, which had the second lowest F1-score (after PHANOTATE), could potentially benefit from switching to using GOODORFS. Since Glimmer is quite modular in its workflow, it is quite easy to change it to use GOODORFS. To facilitate this change, we even tailored our code to mirror the command line arguments and output format of LONGORFS, and have included the code to implement this patch in the GOODORFS github repo. The one lingering shortcoming of GOODORFS is that the current version is coded in Python, which is a scripting language and is significantly slower than compiled code, as is evidenced in our runtime calculations. In order to make GOODORFS more competitive over competing methods, we have begun work on a faster compiled C version.

An area where our GOODORFS method might excel is gene prediction in metagenomes. Most of the available methods for gene prediction in metagenomic reads again rely on reference databases of known genes, which, as previously discussed, has its disadvantages. In contrast, GOODORFS is reference-free; the only prior information needed is the amino acid translation table. Depending on the sequencing technology used, generally a read only contains a fragment of a protein-coding gene, with the beginning or the end (or sometimes both) of the gene extending beyond the length of the read. Because GOODORFS allows for ORF fragments (the lack a start or stop codon) at the edges of the input sequence, it has the ability to work on metagenomic reads. All that is needed is to bin reads according to their GC content and then run the bins through GOODROFS in batches in order to predict gene fragments within the reads. We have already begun adapting and testing the GOODORFS code to work with metagenomes, and instructions are available on the github repo.

## Figures and Tables

**Figure 1 microorganisms-09-00129-f001:**
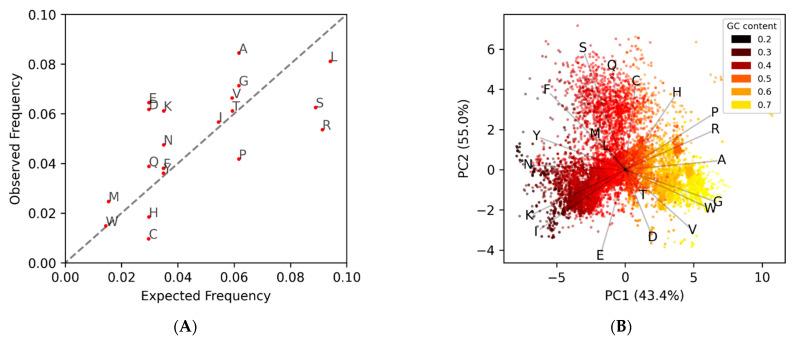
Visualizing the amino acid composition of open reading frames. (**A**) Comparison of average amino acid occurrence across 14,179 phage genomes. Points correspond to the 20 different amino acids and are labeled according to their International Union of Pure and Applied Chemistry (IUPAC) single letter abbreviations. The observed frequencies come from the annotated genome consensus gene calls, while the expected come from the overall codon probabilities calculated from the GC content. Amino acids above the diagonal identity line occur more frequently than expected in coding open reading frames (ORFs), and those below it occur less frequently than expected, which alludes to a coding bias signal. (**B**) The averaged amino acid frequencies of coding ORFs change based on the GC content. The previous consensus calls were averaged for each genome and then plotted using principle component analysis (PCA), and are colored based on the GC content of the genome. The (red) lower GC content genomes tend to favor the amino acids (FYNKI) with AT-rich codons, while the yellow high-GC content genomes tend to favor the amino acids (PRAGW) with GC-rich codons.

**Figure 2 microorganisms-09-00129-f002:**
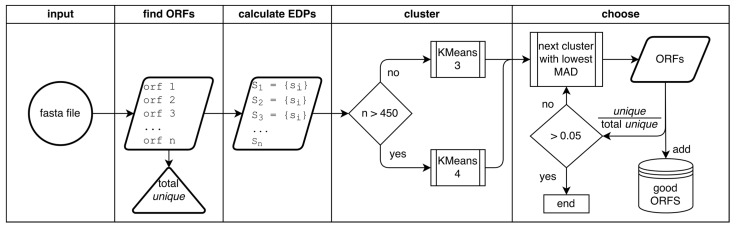
Flowchart of the GOODORFS workflow. After supplying GOODORFS with a fasta file that contains the genome in question, the four major steps are finding the ORFs, calculating the EDPs of the ORFs, clustering the EDPs, and choosing the cluster that contains the good (coding) ORFs.

**Figure 3 microorganisms-09-00129-f003:**
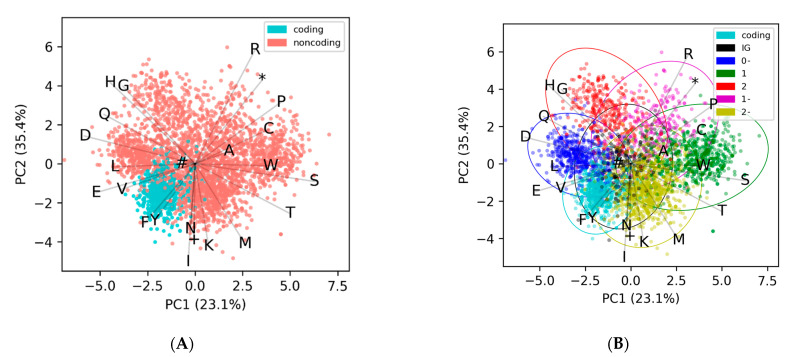
(**A**) The amino acid frequency EDPs of coding and noncoding ORFs for the representative genome *Caulobacter* phage cluster separately. All potential ORFs were found, taking only the longest (i.e., the first outermost available start codon) truncation, finding the amino acid frequencies, coloring them according to whether they are in the consensus annotations (coding), and then plotting them in a PCA. (**B**) The same potential ORFs from the previous figure, except with the noncoding ORFs colored according to their offset in relation to the coding frame (0−, 1, 2, 1−, 2−), or intergenic (IG) if they do not overlap with a coding ORF. The projections for the amino-acids are labeled according to the single letter abbreviations, while the three stop codons amber (+), ochre (#), umber (*) are labeled according to their symbols.

**Figure 4 microorganisms-09-00129-f004:**
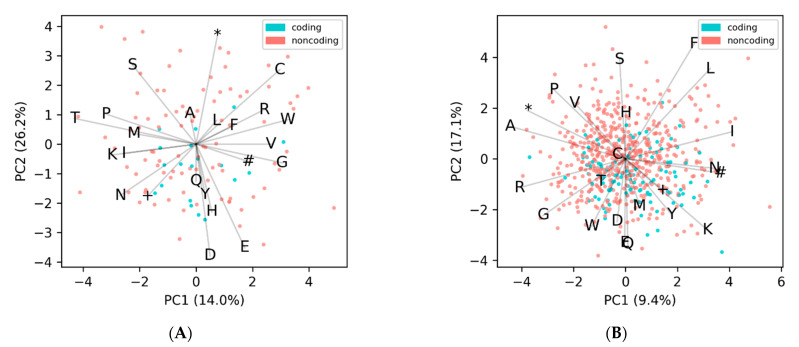
Two examples of phage genomes where the amino acid frequencies of coding and noncoding ORFs do not follow the general trend of clustering separately. Shown here are unique ORFs for (**A**) the filamentous phage *Ralstonia* RSM1 and (**B**) *Escherichia* phage fp01. Other filamentous phages show the same lack of observable coding bias, which could be due to the small genome size; however, it is clear that the *Escherichia* phage does not use the Standard genetic code. The projections for the amino-acids are labeled according to the single letter abbreviations, while the three stop codons amber (+), ochre (#), umber (*) are labeled according to their symbols.

**Figure 5 microorganisms-09-00129-f005:**
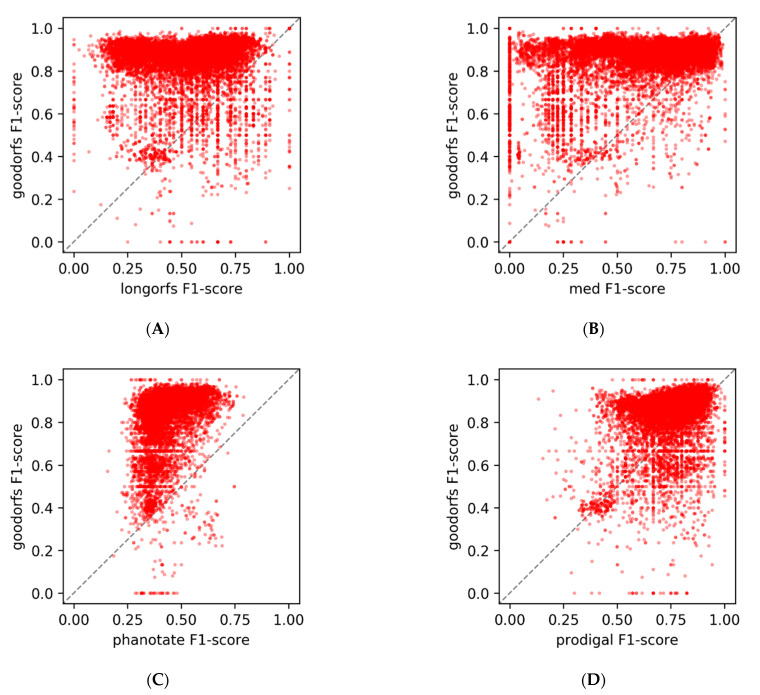
Comparison of the F1 scores for all 14,179 genomes between GOODORFS and (**A**) LONGORFS, (**B**) MED2 and (**C**) PHANOTATE’s training set creation steps. (**D**) Prodigal’s training set creation step. In each panel the dotted line represents x = y, and so points above and to the left of the line represent more accurate protein-encoding gene identification by GOODORFs, while points to the lower/right of the line indicate less accurate protein-encoding gene identification. Points on the line indicate agreement between the algorithms.

**Table 1 microorganisms-09-00129-t001:** Comparison of various performance metrics for GOODORFS and four other similar methods. The listed values are means across all 14,179 genomes, except for the number of genomes failed, which is the number of genomes that had F1-scores lower than 0.1. All values are rounded up to the nearest significant digit.

	GOODORFS	LONGORFS	MED2	* Phanotate	* Prodigal
precision	0.79	0.92	0.60	0.28	0.65
recall	0.90	0.39	0.56	0.96	0.92
accuracy	0.94	0.89	0.87	0.58	0.65
F1-score	0.83	0.53	0.55	0.43	0.75
genomes failed	32	49	1139	0	0
runtime (sec)	93	1	1	11	1

* For PHANOTATE and Prodigal only the initial training set creation step is run, and not the entire gene-finding algorithm.

## Data Availability

The data presented in this study are openly available in FigShare at (10.6084/m9.figshare.13542962.v1), reference number [15].

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
