# Peer review of "Utilizing Amino Acid Composition and Entropy of Potential Open Reading Frames to Identify Protein-Coding Genes"

_microorganisms, 2021, doi:10.3390/microorganisms9010129_

Round 1

Reviewer 1 Report

The manuscript ID: microorganisms-1040650 by McNair et al., entitled “Utilizing amino-acid composition and entropy of potential open reading frames to identify protein-coding genes” presents new bioinformatic tool GOODORFS which may be useful for the identification of protein-coding genes within a set of all possible ORFS.

General remarks:

Dear Authors,

I am always impressed by the creation of a new bioinformatics tool that may outperform the already available ones. There is always room in the field and new paths are waiting to be cleared. The proposed manuscript seems interesting, however, I am unaccustomed to such scarce discussion and few references. Furthermore, according to the microorganisms template, results and discussion sections should be separated. Some information from the introduction could be placed in the discussion section. I encourage you to add, a flowchart describing the procedures/parameters implemented in GOODORFS (and maybe parameters used by the other methods).

In general, I would expect more discussion on the comparison to the other tested methods, applicability and limitations of GOODORFS. Just to encourage potential users who are used to already available tools. Identification of protein-coding genes is difficult and often they can be annotated only as hypothetical proteins. Can you add any discussion on how differentiation of coding from noncoding ORFs may enhance the data analyses?

I am personally interested in the identification of viral sequences from (meta)genomic DNA of bacteria/archaea. Can GOODORFS (or your future tools) somehow contribute to this matter?

Specific remarks are listed below.

Line 26 consider adding precision, recall and accuracy to F1-scores

Line 50 you may remove “((1)” from the equation as it is the only one and there is no reference to this formula in the text

Line 57 use a capital letter for „a” and „b”. alternatively, change in the text (Figure 1a). Change for all figures throughout the manuscript. Figure 1 is prepared based on your analyses, so in my opinion, it should be placed in the result section  

Line 77 Propionibacterium, check bacterial names (with italics) throughout the manuscript

Line 80-81 consider changing to “..is that only genes similar to those already known are predicted.”

Line 88 add a dot before “Currently”

Line 117 consider changing “This was done as the correct stop codon was not known, but the assumption was made that …”

Line 127-128 be consistent, either “unique ORF” and “same ORF” or “unique” ORF and “same” ORF

Line 129 is there any reason to give three references? Does the most recent [14] not include information from [12,13]?

Line 145 Is there any reason for 5% or “just because”?

Line 164 as above, be consistent coding/noncoding or coding/noncoding also check for offset/offset, Microviridae/Microviridae etc.

Line 168 correct the figure, as the legend overlaps with the graph. For example on Figure 2B "P" overlaps with the red rectangle

Line 169 Caulobacter

Line 179 match the names in the main manuscript and supplementary. “SuppFigure 1”, ”Supplementary Figure 1” or just “Figure S1”. Change throughout the manuscript

Line 184 unbold “six”

Line 201 on the SuppFigure 3 in seems that the lowest amount of unique ORFs is approximately 25

Line 211 did you mean Figure 2B?

Line 225 there is no Figure 1C. change to SuppFigure 4.

Line 228-230 any discussion with the literature?

Line 232-240 it would be nice to see these two examples marked on the SuppFigure 4?

Line 249 consider adding additional information about the number of genomes with  F1-score less than 0.1 or maybe also time for analyses.

Line 258 change to “Comparison of the F1-scores for all 14,179 genomes between GOODORFS and A) LONGORTS; B) MED2; C) Phanotate; D) Prodigal…” remove names of the four methods from the top of the figures.

Line 279-280 present full title for Figure S3. It should be “Histograms showing the distribution of genome sizes in (A) the number of genes in the annotation and (B) the number of potential unique ORFs.”

 Line 274-276 I agree that new methodology needs to be further evaluated by case studies but I would personally use different wording to attract potential users.

supplementary figure 3 - what do the numbers on the y-axis refer to? genomes sizes, number of genomes of a certain size? Why there is a different scale? Please clarify, does that mean for example that approximately 1500 genomes of a certain size (?) have 60 of annotated genes?

Supplementary figure 4 - change to “As datasets often do not have full taxonomic lineages, the estimation was made according to the first 8 characters of the sequence name”. I understand the idea of ​​a simplification but maybe it is worth adding information that this classification may underestimate/overestimate some taxa. Also, remove “other” category from the legend as you do not show them.

Reviewer 2 Report

The authors propose a new protein-coding gene predictor (called GOODORF) using the amino-acid composition and the Entropy density profile of the potential open reading frames (ORFs).  

  • Other tools were developed to predict protein-coding genes, but those tools use previous knowledge as a training set.  
  •  GOODORF is distancing itself from the literature tool using an unsupervised machine learning approach using clustering with Kmeans.  
  • They show using F1-measure that GOODORF outperforms the literature tools.  

-- Major revision -- 

I found the paper interesting with a novel approach using unsupervised learning. However, I found the overall structure of the manuscript hard to follow and I believe a significant rewrite would help to readership. For example, the method section requires an important re-structure (probably using subsection including the material, the GOODORF steps and the comparison analysis could help to better organize) and I suggest adding a figure that summarise the step of GOODORF has, including the input format and the output format. That change will help to make better the methods. This is my opinion; buy I seriously think phrases such as “but a future C version is planned for faster runtimes.” are misplaced in a method section (even it is an honest comment but should be better in a “future plan” section). Additionally, a comparative table (version, main machine learning/statistic technique, specific function to do the prediction, limitation and advantages?) with the different tools (MED2, Prodigal, Phanotate, GeneMark, Glimmer, and GOODORF) would be great. 

I have a special issue when the authors refers to PCA plots during the manuscript. It seems that they consider them as a visualization tool. The PCA (principal component analysis) is not a simple visual representation of the dataset. PCA is a reduction method, whose representation it is in a 2D (or sometimes 3D) scatter plot. Moreover, the percentage of variance explained is not shown, and it is crucial to understand the PCA. For example, if the scatterplot representing a PCA whose percentage of variance explained of 10% for the PC1 and 5% for the PC2, provide to the results not conclusive information. Same problem with tSNE plots presented in the supplementary data, in that case, additionally they did not describe it in method or even in supplementary. A tSNE (t-distributed stochastic neighbour embedding) is not either a PCA either a plot representation, but another reduction analysis method as PCA, with also the possibility of plot the results in a 2D or 3D scatter plot. To resume this, It should be better to clarify what is PCA and tSNE in few words to do not confuse the readership.

-- Minor revision-- 

In general, the paper is well-written, but I found some lack of consistency during reading the paper (not very important but necessary to carry out). For example, open reading frames and ORFs are using in both forms. Authors should choose one (I guess that it is better use ORF) and use open reader frames the first time it is referred with the abbreviation between parenthesis: e.g. Line 42 “that are built on the longest (and this most likely to be protein-encoding) open reading frames (ORFs)...”. 

I also found some non-understandable phrases, and some lack or missing justification for some decision: 

  • Line 34, the authors mention different literature tools: GeneMark, Glimmer, Critica, Prodigal and Phanotate. However, Critica is not included in the comparative analysis. As a reader, I would like to know why it was not chosen 
  • Lines 154-155 are not clear for me. It those symbols are then represented in the Figure 2. Then why use the symbology in the main text and not in the figure?  
  • Justification in lines 162 and 163 is not conclusive. They choose an organism because it is the 3rd largest genome, then why does not use the 2nd or the 1st? 
  • In the lines 166 and 167, the authors have the following phrase: “The coding ORFs (blue) and the noncoding ORFs (red) cluster separately, and this same pattern is observed across all other genomes (data not shown)”. Why is the data not shown?  I am interesting to have a supplementary material including a summary information for those organisms 

The paper is finished abruptly. It could be great to have a conclusion to finish the history. In my opinion I would suggest a conclusion section. 

-- Additional point that will improve this paper -- 

In the paper “The effect of sequencing errors on metagenomic gene prediction”, Hoff K. says that ORF-based gene prediction methods are more substantially affected by sequencing errors that cause frameshifts. That could be great if the authors of these papers can considerate that and discuss it.   

GeneMark have a new version (GeneMark-2) that it is not mentioned during the manuscript, I think it is interested to include it in the analysis. Also, a novel tool called AssessORF was published in February 2020, that could be great for the authors to considerate it to compare, but an explanation of why it is not relevant for the study is also valid.  

Hoff KJ. The effect of sequencing errors on metagenomic gene prediction. BMC genomics. 2009 Dec 1;10(1):520. 

[GeneMarks-2] Lomsadze A, Gemayel K, Tang S, Borodovsky M. Modeling leaderless transcription and atypical genes results in more accurate gene prediction in prokaryotes. Genome research. 2018 Jul 1;28(7):1079-89 

Deepank R Korandla, Jacob M Wozniak, Anaamika Campeau, David J Gonzalez, Erik S Wright, AssessORF: combining evolutionary conservation and proteomics to assess prokaryotic gene predictions, Bioinformatics, Volume 36, Issue 4, 15 February 2020, Pages 1022–1029. 

A benchmark analysis to compare different tools could be very interesting, even if GOODORF is not the best because it is developed in Python, thus the phrase ““but a future C version is planned for faster runtimes” could have a better relevance in the discussion.  

At last, a ROC curve analysis in the comparative analysis should be a plus.

Round 2

Reviewer 2 Report

The authors considered the major revision that I suggested for the manuscript. With a better structure in the method section, this is more readable. They provided a new figure representing a flowchart making clearer the GOODORFs framework. Also, they justified why they do not include some other suggestions.